# Associations between Long-Term Air Pollution Exposure and Risk of Osteoporosis-Related Fracture in a Nationwide Cohort Study in South Korea

**DOI:** 10.3390/ijerph19042404

**Published:** 2022-02-19

**Authors:** Seulkee Heo, Honghyok Kim, Sera Kim, Seung-Ah Choe, Garam Byun, Jong-Tae Lee, Michelle L. Bell

**Affiliations:** 1School of the Environment, Yale University, New Haven, CT 06511, USA; honghyok.kim@yale.edu (H.K.); michelle.bell@yale.edu (M.L.B.); 2Interdisciplinary Program in Precision Public Health, Department of Public Health Sciences, Graduate School of Korea University, Seoul 02841, Korea; ssera0905@gmail.com (S.K.); garam0110@gmail.com (G.B.); jtlee@korea.ac.kr (J.-T.L.); 3College of Medicine, Korea University, Seoul 02841, Korea; seungah@korea.ac.kr

**Keywords:** air pollution, bone fracture, cohort, Cox proportional hazard model, osteoporosis, urban environment

## Abstract

Bone health is a major concern for aging populations globally. Osteoporosis and bone mineral density are associated with air pollution, but less is known about the impacts of air pollution on osteoporotic fracture. We aimed to assess the associations between long-term air pollution exposure and risk of osteoporotic fracture in seven large Korean cities. We used Cox proportional hazard models to estimate hazard rations (HRs) of time-varying moving window of past exposures of particulate matter (PM_10_), sulfur dioxide (SO_2_), carbon monoxide (CO), nitrogen dioxide (NO_2_), and ozone (O_3_) for osteoporotic fracture in Korean adults (age ≥ 50 years) in the National Health Insurance Service-National Sample Cohort data, followed 2002 to 2015. HRs were calculated for an interquartile range (IQR) increase. Comorbidity and prescription associated with osteoporosis, age, sex, body mass index, health behaviors, and income were adjusted in the models. Effect modification by age, sex, exercise, and income was examined. We assessed 56,467 participants over 535,481 person-years of follow up. Linear and positive exposure-response associations were found for SO_2_, while PM_10_ and NO_2_ showed nonlinear associations. SO_2_ was associated with osteoporosis-related fracture with marginal significance (HR for an IQR [2 ppb] increase = 1.04, 95% CI: 1.00, 1.09). The SO_2_ HR estimates were robust in analyses applying various moving windows of exposure (from one to three years of past exposure) and two-pollutant models. The central HR estimate of O_3_ implied positive associations but was not significant (HR for 0.007 ppm increase = 1.01, 95% CI: 0.97, 1.06). PM_10_, CO, and NO_2_ did not show associations. Vulnerable groups by sex, age, exercise, and income varied across air pollutants and there was no evidence of effect modifications. Long-term exposure to SO_2_, but not PM_10_, CO, NO_2_ and O_3_, was associated with increased osteoporotic fracture risks in Korean adults.

## 1. Introduction

One of the major public health concerns of aging is bone-related health. The global prevalence of osteoporosis is more than 200 million people, which constitutes 1% of all disability and causes 8.9 million osteoporosis-related fractures every year globally [1]. Osteoporosis can be associated with bone fracture, which can lead to disability or mortality [2]. Studies projected a 2.28-fold increase in the number of osteoporosis-related fractures in the next few decades (1124,060 in 2018 to 2563,488 in 2050) in Asia and a 1.59-fold increase in health care costs (from 9.5 billion US$ in 2018 to 15 billion US$ in 2050) [3,4]. Given that incidence of bone fracture is expected to increase worldwide as the world’s population is rapidly aging, identifying the risk factors of osteoporosis-related fracture is needed for promoting health and well-being in the elder population.

The biological pathways for air pollution’s impact on bone health have been demonstrated and hypothesized in many studies. Air pollution can play a role in bone density and mineralization [5,6]. Substantial studies have suggested putative molecular and cellular mechanisms of bone damage from exposure to air pollution in both animals and humans, which can lead to bone demineralization and osteoporosis [7]. Metals and organic compounds in particulate matter or black carbon can significantly increase serum levels of proinflammatory cytokines (e.g., TNFα, MCP-1, IL-8, MIP-1α, IL-6, IL-1β, receptor activator for nuclear factor κ B ligand (RANKL)), which activate osteoclastogenesis and increase bone resorption by increasing osteoclast precursor [7]. Increase in the levels of IL-6 and TNFα have been reported in mice exposed to sulfur dioxide (SO_2_) [8]. Oxidative stress caused by exposure to ozone (O_3_), nitrogen dioxides (NO_2_), carbon monoxide (CO), SO_2_, and metals in particulate matter (PM) can mediate the cell inflammatory response inducing several proinflammatory cytokines as well. Furthermore, some endocrine-disrupting chemicals in volatile organic compounds or PM may promote cytokines such as IL-17 and induce osteoclasts through the RANKL pathways, whereas the global effect of the mixture of endocrine-disrupting chemicals is unknown [7]. Exposure to SO_2_ may modify levels of bone homeostasis factors such as alkaline phosphatase, magnesium, and calcium, which can lead to osteoporosis, as SO_2_ is associated with sulfur- and oxygen-centered free radicals and cytokine level changes [9,10]. Moreover, outdoor O_3_ can limit UVB photons reaching the ground and reduce the synthesis of vitamin D, a stimulator of the absorption of calcium from the gut, and, therefore, it can lead to bone loss [11].

Numerous studies have suggested that air pollution may be a risk factor of osteoporosis [12,13,14,15] and reduction in bone mineral density (BMD) [12,14,16]. Far less evidence is available for the associations between air pollution and osteoporosis-related fracture. Furthermore, few studies that have examined the impacts of air pollution on osteoporotic fracture show conflicting results. A US study found significant relationships between long-term exposure (one year) to particulate matter with aerodynamic diameter ≤2.5 μm (PM_2.5_) and osteoporosis-related fracture hospitalizations in the Medicare enrollees (aged ≥ 65 years) [14]. A time-series study in Alcorcon, Spain showed significant impacts from short-term exposure to SO_2_, nitric oxide (NO), and NO_2_ on osteoporotic hip fracture incidence [17]. In contrast, a study in Oslo, Norway suggested no associations between self-reported forearm fractures in adults and air pollution (PM_10_, PM_2.5_, NO_2_) [16]. A longitudinal study in Korean found no associations between osteoporotic fracture and PM_2.5_ among women age ≥ 50 years [18]. These discrepancies may be attributable to different study designs, methods for ascertaining outcomes, assessment of exposure period, and capability of adjustment for high risk factors of osteoporosis-related fracture and behavioral patterns. Regarding effect modification by behavioral patterns, the links between air pollution and bone health may differ by exercise. Higher tendency of outdoor activities may increase bone health or expose people to potential risks of bone fracture from injuries. However, the evidence for effect modifications for air pollution and osteoporotic fracture is still lacking.

This study aims to assess the associations between osteoporosis-related bone fracture with long-term exposure to particulate matter (PM_10_), SO_2_, NO_2_, O_3_, and CO in major urban cities in South Korea to examine if persons living in regions with higher air pollution levels have higher risks of bone fracture, using longitudinal data over 10 years. Osteoporotic fracture has become a significant public concern in South Korea, where more than half of women age ≥ 50 years experience osteoporotic bone fractures during their life time [19]. A Korean nationwide study reported that the mortality risk was about three times higher in patients with hip fracture compared to persons without hip fracture [20]. Within one year after vertebral fracture, mortality risk increased 3.5 times for men and 2.5 times for women in South Korea [20]. In addition, studying this population is significant as Asian ethnicity is a vulnerability factor for osteoporosis [21]. Calcium is a critical source for maintaining bone mass in adults, and studies have demonstrated that calcium intake from food was significantly lower in the Korean population compared to the American population based on the national nutrition health study [22]. Information on multiple time-varying measures of air pollution and individual-level confounders of osteoporosis-related fracture in the nationwide large cohort offers a novel opportunity to examine the associations in long-term exposure to air pollution and incidence of osteoporosis-related fracture. This research is unique in its focus on various exposure windows, improved ascertainment of osteoporotic bone fracture, and examination of effect modification by individual- and community-level factors.

## 2. Materials and Methods

### 2.1. Study Population and Study Design

The Korea University Institutional Review Board approved this study. This cohort study used the National Health Insurance Service-National Sample Cohort (NHIS-NSC) data, in which information is available for age, district of address of that year, month of death if occurred during the follow-up period, income level, all prescription drugs, and treatment claim records for enrollees for the period between 2002 and 2015. Based on the Act on Social Security, all Koreans are eligible to become members of this national health insurance system. Enrollment of the national insurance system covers about 97% of the entire population in South Korea [18]. The NHIS-NSC dataset provides the cohort data on 2% of enrollees sampled from the total national insurance enrollees by age, sex, region, insurance type (e.g., employer-sponsored insurance beneficiaries, self-employed insurance beneficiaries, dependent of insurance beneficiaries), and income. This data are an open cohort where participants alive on January 1 of the year following their enrollment in the NHIS remain in the cohort and each participant has varying enrollment year. The NHIS is legally obligated to provide the national health checkup for insured adults. Fully-insured persons through their employment and self-insured head of household can take health checkup regardless of their age, while others in the household (e.g., dependents) can undergo the checkup from age 40 years old or older [23]. Employees working in non-office environments are required to take the health examination every year, and those in other occupations can take it every 2 years [23]. We limited our study regions to the target regions of the capital city (Seoul) and 6 metropolitan cities (Daejeon, Daegu, Incheon, Gwangju, Busan, Ulsan) in South Korea (*n* = 541,253) and targeted persons ≥50 years during the first 3 years of their enrollment in the NHIS-NCS dataset in these target regions (*n* = 84,544). The NHIS-NSC database provides the district of address of the participant for every year during the follow-up. The spatial basis for residential address of the enrollees in the database is district (administrative unit equivalent to the U.S. borough). Follow-up periods were calendar years.

To conduct this longitudinal study, we targeted those who had the health checkup data and lived in our target study regions in the baseline period (i.e., the first 3 years from entry) among adults ≥ 50 years from the NHIS-NSC data. We defined the baseline period as the first 3 years from entry and the follow-up start year as the 4th year from the entry to examine long-term past exposure to air pollution up to the 3 preceding years. We excluded persons who already had diagnosis of osteoporosis-related fracture at the baseline period (*n* = 1526), persons who were lost before the analysis start year (*n* = 16,641), and persons who did not have data of health checkups (*n* = 9910). As a result, about 66.8% (*n* = 56,467) of the persons age ≥ 50 years at the entry year in the target regions were included in the final analysis (*n* = 84,544) (Figure 1). As we excluded participants lost for follow-up during the baseline period (i.e., death, move-out, loss of enrollment for the insurance system), higher percentages of older age groups (i.e., ≥65 y) were found among the excluded participants, but there were no significant differences for income or sex between the included participants and the excluded participants (Appendix A).

We identified the initial (first) occurrence of bone fracture at osteoporosis-related sites (i.e., hip, spine, forearm, humerus [upper arm]) using the operational diagnosis [24,25,26,27] based on the primary diagnosis according to the 10th version of International Classification of Disease (ICD-10) codes that occurred during the period between 2002 and 2015 and based on procedure codes (e.g., open reduction and internal fixation, percutaneous/close pinning, external fixation, cast, skeletal traction, kyphoplastry, etc.). We included fracture in vertebrae (cervical vertebra and other parts of neck [ICD-10: S12]; lumbar spine and pelvis [S32]), forearm (S52), humerus (upper end of humerus [S42.2]; shaft of humerus [S42.3]), and hip (head and neck of femur [S72.0]; pertrochanteric fracture [S72.1]; subtrochanteric fracture [S72.2]) as applied in previous studies [18,20]. Other pathological fractures (ICD-9: 733.1 or ICD-10 M84.4, M84.5, M84.6, M84.7) that were not osteoporotic fractures were excluded. Participants who had diagnosis of osteoporosis-related fracture during the baseline period were excluded.

### 2.2. Air Pollution Assessment

We obtained the monitoring data of hourly PM_10_, SO_2_, CO, NO_2_, and O_3_ concentrations from the National Ambient air Monitoring Information System (NAMIS) for 2001–2015 [28]. PM_2.5_ data were not available because monitoring started in 2013 in our target cities. The system included 133 stations across 74 districts in the 7 target cities and provided verified observation data for 2001–2020. The average number of monitoring stations in each district was 2.4 (minimum = 1, maximum = 7, Q1 = 2, Q3 = 4). The average size of study districts was 73.1 km^2^ (minimum 2.8, maximum 753.3, Q1 = 17.9, Q3 = 68.2). The year 2002 was selected as the start of exposure assessment in this study because that is the first year that the NHIS database provides residential location (district) of the cohort participants. We calculated daily means for each of the 5 air pollutants from hourly observations from each monitoring station. Then, annual mean of air pollution concentrations was calculated by averaging the daily mean values from monitoring stations within each district for each air pollutant. These district-specific annual mean values were used to assess long-term exposure levels for each participant in that district as described in the statistical analysis section.

### 2.3. Covariates

We obtained information from the NHIS-NSC dataset for those who conducted health checkups. Variables included body mass index (BMI), smoking status, alcohol consumption (glasses per week), and frequency of physical exercise per week (0, 1–2, 3–4, 5–6, 7 times). We generated a binary variable for high alcohol intake using the alcohol consumption variable. A unit of alcohol varies from 8 to 10 g alcohol among different countries [19]. This is equivalent to a standard glass of beer (285 mL), a single measure of sprits (30 mL), a medium-sized glass of wine (120 mL), or 1 measure of an aperitif (60 mL) [19]. In this study, total daily alcohol intake was measured by the number of glasses of ‘Soju’, and 1 glass of Soju is considered to have 1 unit of ethanol. Given that the recommended amount suggested by the WHO is <5 units per day for men and <2.5 units per day for women, we defined high alcohol intake as 5 or more units for men and 3 or more units for women [19].

We considered the risk factors in the WHO Fracture Risk Assessment Tool (FRAX) algorithm. The FRAX algorithm provides 10-year absolute fracture risk of a person based on BMI, smoking status, excessive alcohol intake, use of oral glucocorticoids, diagnosis of rheumatoid arthritis, and diagnosis of secondary causes of osteoporosis [19]. We defined exposure to long-term oral glucocorticoids in each person and year if the participant was prescribed oral glucocorticoids (i.e., hydrocortisone, cortisone acetate, prednisone, prednisolone, methylprednisolone, triamcinolone) for more than 30 days in that the prior year [19]. Using all claim data based on ICD-10 codes for each participant during the follow-up period, we examined if a participant had confirmed diagnosis (both hospital visit and hospitalization) of rheumatoid arthritis during the follow-up period. Secondary causes of osteoporosis include disorders strongly associated with osteoporosis. These disorders include type 1 diabetes, hyperparathyroidism, hyperprolactinemia, hypopituitarism, Cushing’s syndrome, chronic obstructive pulmonary disease, chronic renal failure, severe malnutrition, multiple myeloma, and idiopathic hypercalciuria [29,30]. Three secondary causes of osteoporosis including osteogenesis imperfecta, premature menopause, and hypogonadism were not considered in the analysis. The NHIS-NSC did not provide medical claim data on these causes for privacy and data protection issues. We generated a time-varying binary variable for osteoporosis-related conditions and assigned each participant a ‘yes’ for the year they were diagnosed with any of these symptoms and the following years during the follow-up period; otherwise, a ‘no’ was assigned. We also calculated the Charlson Comorbidity Index (CCI) for each participant and each follow-up year by reviewing the claim data in the NHIS cohort database to identify comorbid disease. Details of the included comorbid diseases and calculation methods were adopted from a previous study [31]. We also generated a time-varying binary variable for use of anti-osteoporosis agents (i.e., sodium alendronate, raloxifene hydrochloride, risedronate sodium, raloxifene hydrochloride, teriparatide, teriparatide acetate).

### 2.4. Statistical Analysis

Person-years of follow-up were considered for each participant in our statistical analysis. Follow-up time in the analysis was defined for each participant as the period from 1 January in the analysis start year (i.e., the first year after the baseline period) to the date of whichever occurred first: incidence of osteoporosis-related fracture, the end of follow-up (31 December 2015), death, moving out of the seven cities, or the loss to follow-up. Multivariate Cox proportional hazards models stratified by age (5-year interval) and sex were used to estimate hazard ratios (HRs) of bone fracture and 95% confidence intervals (CIs) in relation to long-term exposure to air pollution for an interquartile range (IQR) increase. For the participants included in the analysis, a right censoring occurred for those known to have moved out of our target cities, deceased, or lost enrollment for the NHIS between 2005 and 2015.

The district-specific average air pollution concentration values were assigned to each participant based on their residential district. To reflect changing levels of PM_2.5_ over time, we considered time-varying variables for air pollution. Moreover, we considered various exposure windows for air pollution in our analyses because the appropriate exposure period for developing osteoporosis-related fracture is unknown. We used 1-, 2-, and 3-year moving annual average concentrations of PM_10_, SO_2_, NO_2_, O_3_, and CO in the prior year as a time-varying variable to estimate bone fracture risk in that year. We also applied a 2-year moving average of air pollution with a 1-year lag for each year (e.g., exposure in 2002–2003 for the fracture risk in 2005). We examined up to a 3-year moving window of past exposure considering the bone remodeling cycle. In normal bone structure, the median duration of the remodeling cycle in cortical bone is 120 days, and the total surface of cancellous bone is completely remodeled over a period of 2 years [32]. Hormonal disorders or some treatments can shorten the remodeling duration to 100 days or increase it to 1000 days [32].

For each air pollutant, we applied two-pollutant models additionally adjusting for air pollutants as the continuous variable to account for mutual correlation of pollutants.

In our main time-dependent models, we controlled for the potential confounding effects of exposure to oral glucocorticoids in the past year (yes, no), diagnosis of rheumatoid arthritis (yes, no), diagnosis of secondary causes of osteoporosis (yes, no), income-based insurance fee (<40th percentile, 40th–79th percentiles, ≥80th percentile; higher percentile indicating higher income), smoking status (current, former, and never), high alcohol consumption (yes, no), BMI, and frequency of exercise per week (0, 1–2, 3–4, 5–6, 7 times). All variables were included as time-varying variables. We verified the proportional hazard assumption of the time-fixed variables (i.e., sex) by plotting log–log Kaplan–Meier survival estimates [33]. We examined potential effect modification by age (50–69 y, 70+ y), sex (men, women), sex and age (men 50–69 y, men 70+ y, women 50–69 y, women 70+ y), frequency of exercise (low: never, moderate: 1–4 times per week, high: 5–7 times per week), and income (low: 0–39th, moderate: 40–79th, high: 80–100th percentile). Effect modification was examined by adding a multiplicative interaction term between air pollution and each effect modifier. Significance of the interaction term was examined by the *p*-value at a significance level of 0.1.

To examine the shape of the exposure–response relationship in this cohort, natural spline functions with 4 degrees of freedom (df) were applied for each air pollutant using Cox models, adjusted for all covariates. In estimating HRs, linear exposure–response relationships were applied in Cox models.

## 3. Results

Characteristics in the entry year or during the follow-up years of the included participants are shown in Table 1. Participants with and without outcomes differed in all characteristics except for diagnosis of secondary causes of osteoporosis during follow-up, which justified adjustment of the covariates. The average time of follow-up was 9.5 years and was higher in persons who did not experience osteoporosis-related fracture (6.4 years). Among the total 56,498 participants, 5398 participants (9.6%) experienced osteoporosis-related fracture. Incidence of these fractures was significantly higher in women; 77.7% of the participants who experienced a fracture were women. About half of participants (48.6%) were age 50–64 years in their cohort entry year. Participants were exposed to PM_10_ levels exceeding the national annual standard (50 μg/m^3^). Long-term exposure concentrations of air pollution for the entire follow-up years for each participant were significantly different between the group experiencing the outcome and the group not experiencing the outcome for all air pollutants.

The levels of PM_10_, SO_2_, NO_2_, and CO were positively correlated, while the levels of O_3_ showed negative correlations with PM_10_, CO, and NO_2_ (Appendix A). The degree of negative correlations of PM_10_, CO, and NO_2_ with O_3_ were moderate (*r* < −0.5).

Linear Cox models showed marginal significance for elevated risk for osteoporosis-related fracture at higher levels of ambient SO_2_ (Table 2). The HR for an IQR increase in SO_2_ concentration (2 ppb) was 1.04 (95% CI: 1.0, 1.08). The HRs and 95% CIs for osteoporosis-related fracture associated with SO_2_ were robust in all examined exposure windows. The HRs for an IQR increase were 1.00 (95% CI: 0.93, 1.07) for PM_10_ (IQR = 12.7 mg/m^3^) and 0.99 (95% CI: 0.94, 1.04) for CO (IQR = 0.192 ppm) in the models using 3-year moving annual average exposure levels. The HR for an IQR increase (0.012 ppm) was 1.01 (95% CI: 0.97, 1.05) for NO_2_ in the model using a two-year moving annual average with a one-year lag. The HR for O_3_ was 1.01 (95% CI: 0.97, 1.06, IQR = 0.007 ppm) in the models using one- and two-year moving annual average levels.

Table 3 shows HRs of osteoporosis-related fracture associated with air pollution by age, sex, frequency of exercise, and income. The impacts of PM_10_, CO, NO_2_, and O_3_ were not significant in any subgroups, while the HRs were significant in some subgroups for SO_2_. The effect modification by frequency of exercise per week was significant at a significance level of 0.1 (*p*-value = 0.064) for PM_10_, but no HRs were significant for the subgroups of exercise. The interaction terms of each effect modifier indicated no difference in the risks. As a result, there were no significant differences in the HRs among subgroups for all air pollutants.

Figure 2 demonstrates that the three-year moving annual average SO_2_ showed nearly linear and positive relationships with osteoporosis-related fracture. Decreasing log HR was found above the national standard of annual mean PM_10_ (i.e., 50 μg/m^3^). The associations for CO showed monotonic curves.

In models including simultaneous exposure to two air pollutants, the HR estimates of air pollution remained robust. The effect estimates of SO_2_ slightly attenuated and remained statistically significant after adjustment for PM_10_ and O_3_ (Appendix A).

## 4. Discussion

Our study found significant and positive associations between long-term exposure to SO_2_ and osteoporosis-related fracture, which is similar to the finding from a previous time-series analysis focusing on short-term air pollution exposure in Spain [17]. We did not find statistically significant associations with PM_10_ exposure. This is similar to the findings from a cohort study in Norway examining PM and self-reported forearm fracture [16]. Similarly, a previous Korean study using the same cohort sample datasets, focusing on women aged >50 y, applying a time-independent PM exposure around the middle point of the follow-up period (2008–2009), and adjusting for age, income, CCI, BMI, smoking, drinking, exercise, and total cholesterol, did not find associations of PM_10_ and osteoporotic fracture [18]. According to a recent systematic review [34], O_3_ and hip fracture has only been studied in a time-series analysis conducted for Spain [17], and this study showed no associations between O_3_ and hip fracture. While our study was based on retrospective cohort design and examined long-term air pollution exposure, we also found no associations between O_3_ and osteoporotic fracture. The finding of no associations between NO_2_ and osteoporotic fracture in our study was similar to results in the Norwegian cohort study showing the impact of long-term NO_2_ exposure on self-reported forearm fracture [16], but it differed from the significant association between short-term NO_2_ exposure and hip fracture in Spain [17]. While these varying study results warrant future research, our cohort study adds evidence on the associations between long-term air pollution exposure and osteoporosis-related fracture in adults.

Research gaps also remain regarding what is the appropriate exposure period for developing osteoporosis-related fracture. Currently, there is no consensus for windows of long-term exposure to air pollution in relation to osteoporosis-related fracture risk. The exposure windows used in the studies for bone fracture also varied. Alver et al. applied a time-independent variable for air pollution exposure (PM, NO_2_) from 1992 to 2001 in their cohort study for self-reported forearm fracture [16]. Chiu et al. used average PM_2.5_ concentration for the one year before the index date in their case-control study of osteoporotic fracture [1]. Prada et al. assessed the association of one-year PM_2.5_ average and the count of hospital admissions from osteoporotic fracture for each year in 2003–2010 using generalized linear mixed models [14]. Although we examined past exposure up to three years, we were unable to assess exposure from the more distant past. It is unknown if the impacts from air pollutants may persist for a longer period, which warrants future research.

We found significant associations between SO_2_ exposure and osteoporosis-related fracture in Korean adults. This finding can be reinforced by the results from a recent study in Taiwan reporting that SO_2_ exposure was associated with increased osteoporosis risks in adults [8]. However, less is known regarding mechanisms for SO_2_ compared to other air pollutants such as PM, O_3_, or NO_2_, which warrants more work on the physiological mechanisms for air pollution including SO_2_. Large quantities of SO_2_ are emitted from fossil-fuel burning (e.g., coal, oil, diesel) and other industrial activities containing sulfur. Moreover, SO_2_ and resulting pollutants can travel large distances far from the emission sources affecting a large number of residents [35]. SO_2_ is generally highly correlated with other air pollutants, especially NO_2_ and PM, which were positively correlated with osteoporosis in previous studies [36]. During the study period, high SO_2_ emission and SO_2_ concentrations were consistently observed in four of our seven target cities (e.g., Daejeon, Ulsan, Incheon, Busan) in our study, which contain many fossil-fuel-burning plants located at seaports and industrial complexes [37,38]. Although focusing on different health outcomes (e.g., mortality), a previous study using the Korean national health and nutritional examination surveys with mortality follow-up also found a similar pattern that associations between SO_2_ and all-cause or cardiovascular mortality were larger than for the associations for NO_2_ or PM_10_ in seven metropolitan cities in South Korea [37]. Another Korean study based on mortality in these cities found that the association between short-term PM_10_ exposure and all-cause or cardiovascular mortality was higher in cities with a higher ratio of SO_2_ level to PM_10_ level [39]. These results suggest that SO_2_ may be a proxy for other air pollutants (e.g., finer particles, sulfate) that are not fully captured by PM_10_. We also identified an association between SO_2_ exposure and osteoporotic fracture that persisted with adjustment for air pollution in the two-exposure models.

We did not find associations between O_3_ and osteoporosis-related fracture in our analyses, which is aligned with a result by a time-series study in Spain by Mazzucchelli showing no associations between short-term O_3_ exposure and osteoporotic fracture [17]. O_3_ showed high correlation with PM_10_, CO, and NO_2_ in our study cities indicating the difficulty in separating the influence of O_3_ from that of the other air pollutants. The reduction of O_3_ HR estimates in the two-pollutant models supports this as well. On the other hand, our study using the fixed-location measurements did not have information of spatial patterns of O_3_ within a district. Therefore, high-resolution O_3_ modeling data would be useful for assessing the association between long-term O_3_ exposure and osteoporotic fracture with decreased exposure errors. Misclassification for NO_2_ exposure could also be relevant for the null associations between NO_2_ and osteoporotic fracture in this research. Moreover, NO_2_ reacts and decays more quickly compared with PM_2.5_ [40]. Uncertainties remain regarding how the exposure errors from these sources would affect the associations between NO_2_ and osteoporotic fracture in our study regions. Like O_3_, high-resolution data of NO_2_ would be useful in future studies.

Our study showed significant associations between long-term SO_2_ exposure and osteoporosis-related fracture in women, whereas no significant associations were not found for men. Higher risks of osteoporosis in female adults than male adults have been suggested in previous studies [41]. Estrogen deficiency in postmenopausal females is a known risk factor for osteoporosis and osteoporotic bone fracture [1]. We also found significant associations between SO_2_ and osteoporosis-related fracture in the younger age group (50–69 years), high income persons, and persons with higher frequency of exercise. However, a caution is needed in determining the effect modifications for these subgroups. The percentage of persons aged 50–69 years in the entry year was 83.9%, and the low percentage of persons ≥70 years would have affected the statistical power of the subgroup analysis. The percentage of persons who exercised ≥5 times per week was 71.6%, and the percentage of the high-income group (80–100th percentile of the income distribution) was 46.5%. Other possible reasons for the higher association for persons with higher frequency of exercise include potentially higher personal exposure to ambient air pollution during exercise outdoors and higher ventilation rates while exercising, although the data does not address the type of exercise or location of exercise.

The HRs of air pollution and all considered covariates are shown in Appendix A. Significant associations with osteoporosis-related fracture were found for smoking (HR = 1.24 for current smokers, 95% CI: 1.10, 1.40) and high alcohol intake (HR = 1.27, 95% CI: 1.10, 1.52), CCI (HR = 1.02, 95% CI: 1.01, 1.03), which were consistent with previous studies [19,42]. Moreover, significant HRs of exposure to oral glucocorticoids (HR = 1.28, 95% CI: 1.13, 1.46) and a 10 kg/m^2^ increase in BMI (HR = 0.85, 95% CI: 0.77, 0.93) showed consistent results with previous studies [19].

Our study has some strengths. To our best knowledge, this is the first study to assess the impact of long-term SO_2_ and CO exposure in relation to osteoporotic fracture. This is also the first longitudinal study to adjust for secondary causes highly associated with osteoporosis (i.e., factors considered in FRAX) in assessing the associations between air pollution exposure and osteoporosis-related fracture. Various moving windows of exposure to air pollution were examined. We identified osteoporosis-related bone fracture using operational ascertainment based on procedure codes as well as disease codes, (ICD) although we did not have information of osteoporosis based on BMD or laboratory examinations. Previous studies mainly identified osteoporotic fracture using ICD codes [14,17,18], but one study identified patients using ICD codes and surgical codes [1]. Moreover, the cohort in this study is a sample representing a large portion of the entire population. To the best of our knowledge this study is the first to examine effect modification by income and exercise levels for the air pollution impacts on osteoporosis-related fracture. Lastly, we contribute evidence on the air pollution impacts for Asian persons, and this race is known to be more vulnerable to risk of bone fracture [21].

We also have limitations. First, we did not consider temporal changes in chemical compositions of particulate matter, while different chemicals may be differently associated with BMD [7]. Second, use of station-based monitoring data of air pollution and lack of information on individual’s time-activity patterns limit investigation of the spatial heterogeneity of air pollution, the differences between indoor and outdoor concentrations, and varying activity patterns that affect personal exposure. The spatial resolution of this research could obscure within-city heterogeneity of pollution levels as the Korean NHIS-NSC dataset only provides the address information at the district level. Third, we examined PM_10_ rather than PM_2.5_ although this smaller size of particulate matter could also be associated with these health outcomes. Air pollution monitoring data on PM_2.5_ is not available for our full study period and study locations. Future work should investigate this pollutant including comparison of effect estimates from PM_2.5_ and PM_10_ along with consideration of different chemical structures of particulate matter. Fourth, we did not consider the timing of bone fracture within a year due to higher computational burden of person-day or person-month follow-up estimations. Incidence of bone fracture, particularly hip fracture, in the elderly is generally higher in cold seasons, and this seasonality has been suggested to be associated with deficiency in vitamin D or meteorological factors such as cold temperature, rain, ice, and snow [43,44,45], and although our study used annual averages, issues of seasonality warrant further study. Fifth, we also did not have information on BMD, so we did not consider ascertainment of osteoporotic fracture based on BMD. Dietary habits of study populations were not available as well. Sixth, we focused on adults living in urban regions to keep compatibility among participants for characteristics of air pollution exposure and unmeasured contextual confounders. Lastly, we limited our analyses to the cohort participants who had the health examination data. Limiting analysis to subjects with the health checkup data in the analysis has the advantage that various health-related individual-level variables (e.g., smoking, alcohol drinking, exercise) could be controlled, but statistical power would decrease due to lower sample size [23]. Furthermore, the national health examinations are available for fully-insured employees, self-insured heads of household, and dependents >40 y, so the included participants in our study may have different characteristics from the participants who did not have the health examination data in the NHIS-NSC dataset. According to the existing scientific literature, the factors increasing the likelihood of receiving health examinations include younger age, male sex, higher education and income, smoking, higher frequency of exercise, and lower depressive symptoms [46]. Given these patterns, our results may not be generalizable for the entire adult population >50 years in South Korea.

## 5. Conclusions

Long-term exposure to SO_2_ was associated with a significant increase in osteoporosis-related fracture incidence in adults (age ≥ 50 y) in a large cohort with detailed information on medical records and health behaviors. These results were robust to various modeling approaches for exposure window and adjustment for multi-pollutant exposure. The associations varied by frequency of exercise and income, but there were no significant effect modifications by these factors across the studied air pollutants. Our results suggest that long-term exposure to air pollution may be an emerging risk factor of osteoporosis-related fracture in urban populations and imply the importance of prevention of osteoporosis-related fracture risks from air pollution exposure.

## Figures and Tables

**Figure 1 ijerph-19-02404-f001:**
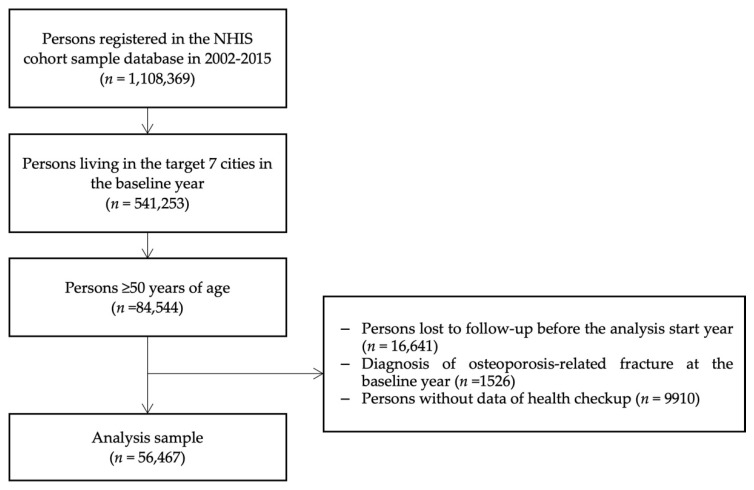
Flow chart of study population in the NHIS-NSC, 2002–2015.

**Figure 2 ijerph-19-02404-f002:**
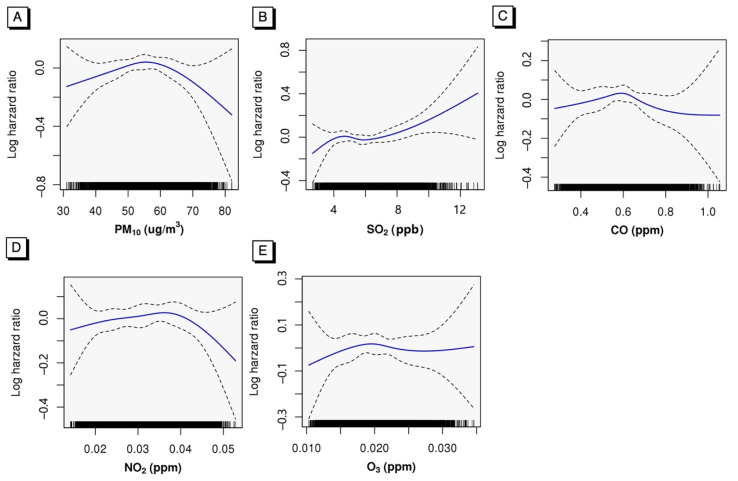
Log hazard ratio of incidence of osteoporosis-related fracture as a nonlinear curve (solid line) and 95% CIs (dashed lines) of air pollution exposure in the Korean National Health Insurance cohort of adults (age ≥50 y) for (**A**) PM_10_, (**B**) SO_2_, (**C**) CO, (**D**) NO_2_, and (**E**) O_3_. The curves were estimated with Cox proportional hazards models using natural splines with 4 df for PM_10_, SO_2_, CO, and NO_2_, and 3 df for O_3_. Models adjusted for age, sex, history of rheumatoid arthritis or secondary causes of osteoporosis during the follow-up period, exposure to oral glucocorticoids in the prior year, use of anti-osteoporosis agents, the Charlson Comorbidity Index, household income-based insurance fee, BMI, smoking status, high alcohol intake, and frequency of exercise per week.

**Table 1 ijerph-19-02404-t001:** Characteristics of participants and air pollution exposure (2002–2015).

		Osteoporosis-Related Fracture	
	Total Participants (*n* = 56,498)	No Event (*n* = 51,100)	Event (*n* = 5398)
Entry year			
Sex (*n*, %)			
Men	26,243 (46.5)	25,041 (49.0)	1202 (22.3) *
Women	30,224 (53.5)	26,030 (51.0)	4194 (77.7)
Age in the entry year (*n*, %)			
50–59 y	20,673 (36.6)	19,301 (37.8)	1372 (25.4) *
60–64 y	15,183 (26.9)	13,929 (27.3)	1254 (23.2)
65–69 y	11,510 (20.4)	10,235 (20.0)	1275 (23.6)
70–74 y	5851 (10.4)	5001 (9.8)	850 (15.8)
75–79 y	2449 (4.3)	1987 (3.9)	462 (8.6)
80–84 y	720 (1.3)	551 (1.1)	169 (3.1)
85+ y	81 (0.1)	67 (0.1)	14 (0.3)
During follow-up			
Charlson Comorbidity Index in the first year of follow-up (mean ± SD)	0.36 ± 0.82	0.35 ± 0.82	0.44 ± 0.86 *
BMI (mean ± SD)	24.2 ± 3.1	24.2 ± 3.0	24.1 ± 3.2 *
Smoking status (*n*, %)			
Never-smoker	39,255 (69.8)	34,818 (68.4)	4437 (82.8) *
Former smoker	9593 (17.1)	9145 (18.0)	448 (8.4)
Current smoker	7399 (13.2)	6925 (13.6)	474 (8.8)
Frequency of exercise per week (*n*, %)			
0 time	9488 (16.9)	8272 (16.3)	1216 (22.7) *
1–2 times	4146 (7.4)	3718 (7.3)	428 (8.0)
3–4 times	2331 (4.1)	2109 (4.1)	222 (4.1)
5–6 times	736 (1.3)	650 (1.3)	86 (1.6)
7 times or more	39,551 (70.3)	36,140 (71.0)	3411 (63.6)
High alcohol intake			
Yes	2106 (4.2)	1965 (4.4)	141 (3.0) *
No	47,560 (95.8)	43,071 (95.6)	4489 (97.0)
Income-based insurance fee in the entry year (*n*, %)			
0–19th percentiles	4644 (8.4)	4150 (8.3)	494 (9.4) *
20–39th percentiles	7454 (13.5)	6751 (13.5)	703 (13.4)
40–59th percentiles	9175 (16.6)	8292 (16.6)	883 (16.8)
60–79th percentiles	9923 (18.0)	8925 (17.9)	998 (19.0)
80–100th percentiles	24,003 (43.5)	21,826 (43.7)	2177 (41.4)
Follow-up years (mean, SD)	9.5 (3.0)	9.8 (2.8)	6.4 (3.1) *
Diagnosis of rheumatoid arthritis during follow-up (*n*, %)			
Yes	5913 (10.5)	5251 (10.3)	662 (12.3) *
No	50,554 (89.5)	45,820 (89.7)	4734 (87.7)
Diagnosis of secondary causes of osteoporosis during follow-up (*n*, %)			
Yes	7733 (13.7)	6997 (13.7)	736 (13.6)
No	48,732 (86.3)	44,074 (86.3)	4660 (86.4)
Exposure to oral glucocorticoids during follow-up (*n*, %)			
Yes	9197 (16.3)	8432 (16.5)	765 (14.2) *
No	47,270 (83.7)	42,639 (83.5)	4631 (85.8)
Air pollution exposure in the last 3 years before the index year (mean ± SD)			
PM_10_ (μg/m^3^)	48.7 ± 7.3	48.2 ± 7.0	53.6 ± 7.9 *
SO_2_ (ppb)	5.67 ± 1.37	5.66 ± 1.36	5.80 ± 1.47 *
CO (ppm)	0.553 ± 0.119	0.549 ± 0.118	0.592 ± 0.121 *
NO_2_ (ppb)	30.3 ± 8.2	30.3 ± 8.2	30.9 ± 7.9 *
O_3_ (ppb)	22.3 ± 4.5	22.5 ± 4.4	20.4 ± 4.1 *
Air pollution exposure in the last 3 years before the index year (*n*, %)			
PM_10_, Q1 (31.1–44.3 μg/m^3^)	14,882 (26.4)	14,298 (28.0)	584 (10.8) *
PM_10_, Q2 (44.4–47.3 μg/m^3^)	14,076 (24.9)	13,385 (26.2)	691 (12.8)
PM_10_, Q3 (47.4–51.2 μg/m^3^)	13,752 (24.4)	12,755 (25.0)	997 (18.5)
PM_10_, Q4 (51.3–82.0 μg/m^3^)	13,757 (24.4)	10,633 (20.8)	3124 (57.9)
SO_2_, Q1 (2.7–4.8 ppb)	14,130 (25.0)	12,566 (24.6)	1564 (29.0) *
SO_2_, Q2 (4.9–5.3 ppb)	14,777 (26.2)	13,868 (27.2)	909 (16.8)
SO_2_, Q3 (5.4–6.3 ppb)	13,870 (24.6)	12,423 (24.3)	1447 (26.8)
SO_2_, Q4 (6.4–13.1 ppb)	13,690 (24.2)	12,214 (23.9)	1476 (27.4) *
CO, Q1 (0.276–0.470 ppm)	14,295 (25.3)	13,426 (26.3)	869 (16.1)
CO, Q2 (0.271–0.568 ppm)	14,318 (25.4)	13,113 (25.7)	1205 (22.3)
CO, Q3 (0.569–0.630 ppm)	13,745 (24.3)	12,318 (24.1)	1427 (26.4)
CO, Q4 (0.631–1.055 ppm)	14,109 (25.0)	12,214 (23.9)	1895 (35.1)
NO_2_, Q1 (14.2–23.5 ppb)	14,642 (25.9)	13,415 (26.3)	1227 (22.7) *
NO_2_, Q2 (23.6–29.9 ppb)	14,269 (25.3)	12,944 (25.3)	1325 (24.6)
NO_2_, Q3 (30.0–36.5 ppb)	13,439 (23.8)	11,992 (23.5)	1447 (26.8)
NO_2_, Q4 (36.6–51.3 ppb)	14,117 (25.0)	12,720 (24.9)	1397 (25.9)
O_3_, Q1 (10.6–19.1 ppb)	14,559 (25.8)	12,501 (24.5)	2058 (38.1) *
O_3_, Q2 (19.2–22.1 ppb)	13,703 (24.3)	11,999 (23.5)	1704 (31.6)
O_3_, Q3 (22.2–25.5 ppb)	14,130 (25.0)	13,038 (25.5)	1092 (20.2)
O_3_, Q4 (25.6–34.6 ppb)	14,075 (24.9)	13,533 (26.5)	542 (10.0)

Note: * Significantly different at a significance level of 0.05 (*t*-test or chi-square test).

**Table 2 ijerph-19-02404-t002:** Cox model hazard ratios (and 95% CIs) for an interquartile range (IQR) increase in exposure to air pollution for osteoporosis-related fracture incidence, by air pollutant (*n* = 56,467).

Exposure (IQR)	HR (95% CI)			
	1-yr Moving Average with No Lag	2-yr Moving Average with No Lag	3-yr Moving Average with No Lag	2-yr Moving Average with a 1-yr Lag
PM_10_ (13.7μg/m^3^)	0.99 (0.92, 1.05)	0.98 (0.92, 1.05)	1.00 (0.93, 1.07)	1.01 (0.944, 1.07)
SO_2_ (2 ppb)	1.04 (1.00, 1.08)	1.04 (1.00, 1.08)	1.04 (1.00, 1.09)	1.04 (1.00, 1.08)
CO (0.192 ppm)	0.99 (0.94, 1.03)	0.99 (0.94, 1.03)	0.99 (0.94, 1.04)	0.99 (0.95, 1.04)
NO_2_ (0.012 ppm)	0.99 (0.95, 1.03)	0.99 (0.95, 1.04)	1.00 (0.96, 1.05)	1.01 (0.97, 1.05)
O_3_ (0.007 ppm)	1.01 (0.97, 1.06)	1.01 (0.96, 1.06)	1.00 (0.95, 1.06)	1.00 (0.95, 1.05)

Note: All models adjusted for age, sex, diagnosis of rheumatoid arthritis or secondary causes of osteoporosis during the follow-up period, exposure to oral glucocorticoids in the prior year, use of anti-osteoporosis agents, the Charlson Comorbidity Index, household income-based insurance fee, BMI, smoking status, high alcohol intake, and frequency of exercise per week.

**Table 3 ijerph-19-02404-t003:** Effect modification by sex, age, exercise, greenspace, and income for the Hazard Ratios (HRs) of osteoporosis-related fracture associations with an IQR increase in three-year moving annual average concentration of air pollution (*n* = 56,467).

	PM_10_ (IQR = 13.7 μg/m^3^)		SO_2_ (IQR = 2 ppb)		CO (IQR = 0.192 ppm)		NO_2_ (IQR = 0.012 ppm)		O_3_ (IQR = 0.007 ppm)	
	HR (95% CI)	*p*-Int.	HR (95% CI)	*p*-Int.	HR (95% CI)	*p*-Int.	HR (95% CI)	*p*-Int.	HR (95% CI)	*p*-Int.
Sex										
Men	0.99 (0.89, 1.11)	0.884	1.00 (0.92, 1.09)	0.300	0.93 (0.85, 1.02)	0.142	1.00 (0.91, 1.09)	0.929	1.03 (0.93, 1.13)	0.614
Women	1.00 (0.93, 1.08)		1.06 (1.01, 1.11) **		1.01 (0.95, 1.07)		1.00 (0.95, 1.06)		1.00 (0.94, 1.06)	
Age (years)										
50–69 y	1.02 (0.94, 1.11)	0.300	1.06 (1.00, 1.11) *	0.441	1.01 (0.95, 1.07)	0.395	1.02 (0.96, 1.08)	0.437	0.99 (0.93, 1.06)	0.650
≥70 y	0.96 (0.88, 1.06)		1.02 (0.96, 1.09)		0.97 (0.90, 1.04)		0.98 (0.91, 1.05)		1.02 (0.94, 1.10)	
Sex and age (years)										
Men, age 50–69 y	1.02 (0.89, 1.17)	0.753	0.96 (0.86, 1.07)	0.130	0.94 (0.84, 1.05)	0.372	0.95 (0.85, 1.07)	0.260	1.02 (0.90, 1.15)	0.940
Men, age ≥70 y	0.93 (0.77, 1.13)		1.08 (0.94, 1.24)		0.92 (0.79, 1.08)		1.08 (0.93, 1.25)		1.03 (0.88, 1.21)	
Women, age 50–69 y	1.02 (0.93, 1.11)		1.09 (1.03, 1.16) **		1.03 (0.96, 1.11)		1.04 (0.97, 1.11)		0.99 (0.91, 1.06)	
Women, age ≥70 y	0.97 (0.88, 1.08)		1.01 (0.94, 1.08)		0.98 (0.90, 1.06)		0.95 (0.88, 1.03)		1.01 (0.93, 1.10)	
Frequency of exercise per week										
Never	0.93 (0.83, 1.04)	0.064	1.04 (0.96, 1.12)	0.980	0.94 (0.86, 1.03)	0.446	0.98 (0.89, 1.07)	0.803	1.02 (0.93, 1.13)	0.900
1–4 times	0.92 (0.79, 1.06)		1.04 (0.92, 1.18)		1.01 (0.89, 1.15)		1.02 (0.90, 1.15)		0.99 (0.87, 1.13)	
5 times or more	1.05 (0.97, 1.13)		1.05 (0.99, 1.10) *		1.01 (0.95, 1.07)		1.01 (0.95, 1.07)		1.00 (0.94, 1.07)	
Income-based insurance fee										
0–39th percentile	1.02 (0.91, 1.14)	0.177	1.03 (0.95, 1.12)	0.429	1.03 (0.94, 1.14)	0.395	1.02 (0.93, 1.13)	0.617	1.01 (0.91, 1.12)	0.611
40–79th percentile	0.93 (0.84, 1.03)		1.01 (0.94, 1.09)		1.01 (0.93, 1.09)		0.97 (0.89, 1.05)		1.04 (0.95, 1.14)	
80–100th percentile	1.04 (0.95, 1.13)		1.07 (1.01, 1.14) **		0.96 (0.89, 1.03)		1.01 (0.94, 1.08)		0.98 (0.91, 1.06)	

*: significant at a significance level of 0.10. **: significant at a significance level of 0.05. *p*-int: *p*-value for interaction.

## Data Availability

Restrictions apply to the availability of these data. Data was obtained from the National Health Insurance Sharing Service and are available at https://nhiss.nhis.or.kr/ (accessed on 22 December 2021) with the permission of the National Health Insurance Service.

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
