# Peer review of "Associations between Long-Term Air Pollution Exposure and Risk of Osteoporosis-Related Fracture in a Nationwide Cohort Study in South Korea"

_ijerph, 2022, doi:10.3390/ijerph19042404_

Round 1

Reviewer 1 Report

please see the attached files

Author Response

Reviewer 1

  1. Introduction: fair
  2. Methods

(1) Figure 2. It’s very unclear and hard to read. Please remove the unnecessary

information and upload a new one with higher resolutions.

A: We noticed that the original figure did not upload property for during the online manuscript uploading process and showed some broken characters and mislocated arrows. We revised the figure so that it shows the graphic properly. We apologize for this issue.

(2) Line 209: “Anti-osteoporosis agent” is preferred and academically used

instead of “medications used for increasing BMD”, since these agents did

not necessarily increase the BMD.

A: Thank you for this suggestion. We changed the wording ‘medications used for increasing BMD’ to ‘anti-osteoporosis agent’ throughout the manuscript.

(3) Did you consider the seasonal effect on air pollution and the osteoporotic fracture since both of them occurred more often in the winter? How did you handle this?

A: We agree that consideration of seasonality is important. We address this issue on several ways. In the original manuscript, we discussed this important point and provided rationales for not directly addressing the seasonality of air pollution and bone fracture in our discussion section and we maintained this text (line 465): “Third, we did not consider the timing of bone fracture within a year due to higher computational burden of person-day or person-month follow-up estimations. Incidence of bone fracture, particularly hip fracture, in the elderly is generally higher in cold seasons and this seasonality has been suggested to be associated with deficiency in vitamin D or meteorological factors such as cold temperature, rain, ice, and snow [40–42], and although our study used annual averages, issues of seasonality warrant further study.” Second, as we used the Cox models based on person-years of follow-up, the seasonality of exposure and outcome was removed in the models. We added new text in the Methods to note that our model was based on person-year estimation to better clarify this issue (Page 5 line 221).

(4) Line 220. You mentioned that those known to have moved out of the target cities would be categorized as censoring. How did you determine this condition? As you said, this database is a nationwide insurance database. I don’t think insurance status can differentiate the place where one actually stayed since people can change residence freely. This issue is very critical to the study design since you only focus on seven cities.

A: We agree that residential mobility is important for these studies given that our exposure levels can vary by location. Fortunately, our database does in fact have this information. The National Health Insurance Service-National Sample Cohort database provides the district of address of the participant for every year during the follow-up. Thus, we could identify if a participant moved to a new address out of the target city during the follow-up year. Loss of insurance status, which can happen when a participant passes away (the database also provides this information with a separate variable), moves to another country, or loses their citizenship, was not used to differentiate the district of address in the analysis. We added text to clarify that our database does account for residential mobility to the Methods section (page 3, line 133).

  1. Results

(1) Line 271-272. These two sentences are misinterpreted. Higher levels of SO2 were not significantly associated with an increased risk of osteoporosis-related fracture since the 95% CI covered the 1.0. Please correct this.

A: We corrected the sentence to “Linear Cox models showed marginal significance for elevated risk for osteoporosis-related fracture at higher levels of ambient SO2”(lines 287-289). We also revised the sentence in the Abstract to “SO2 was associated with osteoporosis-related fracture with marginal significance” (lines 25-26).

(2) Table 3. Please rearrange the row. The current style is hard to read. Line 287-295. What’s the significance level for interaction? You should mention is in the section of method (Line 246). As I know, the p for interaction means whether the outcomes differ between subgroups by modifiers. If all P for interaction did not reach the significance level (usually set as 0.1), it is meaningless to discuss the details

within subgroups.

A: We found that the layout of Table 3 did not upload properly during the uploading process or generating a file for reviewers. We apologize for this issue. We revised Table 3 so that it appears clearly. We added new text for the significance level of the effect modification at line 261. We omitted lines reporting HRs in subgroups for SO2 (lines 305-308) in the Results. We added new text to that the interaction term of exercise for PM10 was significant at a significance level of 0.1, but the HRs of subgroups were not significant (lines 310-312).

  1. Discussion and conclusion: No comments.

A: Thank you for reviewing our manuscript. 

Reviewer 2 Report

This is a nicely written article describing an investigation of the relationship between long-term concentrations of several ambient air pollutants and osteoporosis-related fractures in a cohort in seven South Korean cities.

Major Comments:

  1. My largest concern is with the rationale and interpretation of the result that there is a connection between SO2 exposure and fracture, but not any of the other pollutants. The suggested biological mechanisms (lines 47-63) are provided for PM, O3, NO2, and CO but not for SO2. The authors discuss this somewhat in lines 354-372, but can they assess whether this is a result of potential residual confounding?
  2. The lack of PM2.5 measurements is a major limitation (and may be tied to the findings with SO2 and no other pollutants) The authors state that widespread monitoring of PM2.5 did not occur until 2013. Is there data available in a subset of the cities prior to this time that would enable some evaluation of PM2.5 exposure?
  3. The exact time period for the analyses is inconsistently described. If 2015 is end of follow-up (Line 110 and 216), then why not include exposure information through at least 2014? Otherwise the different exposure period choices (1-year no lag, 2-year no lag, 3-year no lag, and 2-year with 1 year lag) will have different samples. For example, how is the 1-year no lag exposure calculated with 2015 follow-up data if exposure data are only used through 2013 and not 2014? This may just be an issue of inconsistent description, but if it is in fact an issue of different samples then I suggest the analyses be re-run all using the same dataset (i.e., expand the time range of exposure data collected).
  4. Additional information on the exposure assessment would be helpful for understanding the potential for exposure measurement error. How large (km^2) is the typical district? How many monitors are in the typical district?
  5. None of the effect modification comparisons in Table 3 are statistically significant, so I don’t think the authors should highlight specific estimates from that analysis as statistically significant. In other words, since “there was no significant differences in the HRs among subgroups for all air pollutants” (Lines 294-295), then don’t say that the differences are significant: “estimated impact of exposure to 3-year moving annual average SO2 was significant in women...those age 50-69y…women aged 50-69y..” (lines 289-292).

Minor/Editorial

  1. What exposure window is used in the Table S3 analyses? And Tables S4 to S8?
  2. I suggest presenting the justification for 3-year window (Lines 346-350) earlier in the manuscript when first presenting the choice of exposure periods.
  3. In the abstract, what is “central” HR?
  4. Line 28 in abstract: subscript O3
  5. Line 113: NHIS not NIHS
  6. Line 176: “frequency of physical exercise per week” is in days?
  7. I suggest being consistent about ppm or ppb for SO2.
  8. Line 388 has a double negative.
  9. Table 3 is difficult to read. The horizontal lines are inconsistent and the p-values are split onto multiple lines.

Author Response

Reviewer 2

This is a nicely written article describing an investigation of the relationship between long-term concentrations of several ambient air pollutants and osteoporosis-related fractures in a cohort in seven South Korean cities.

Major Comments:

  1. My largest concern is with the rationale and interpretation of the result that there is a connection between SO2 exposure and fracture, but not any of the other pollutants. The suggested biological mechanisms (lines 47-63) are provided for PM, O3, NO2, and CO but not for SO2. The authors discuss this somewhat in lines 354-372, but can they assess whether this is a result of potential residual confounding?

A: We agree that less in vivo evidence for SO2 than the other air pollutants can be a concern. However, the effects on cytokines and oxidative stress of air pollution exposure involve SO2. We added new text for the new potential mechanisms for SO2 and bone health in the Introduction (lines 64). The newly suggested mechanisms indicate that bone mineral density can be affected by exposure to SO2. To address potential confounding from other air pollutants, we applied two-pollutant models as SO2 generally is highly correlated with NO2 or PM (line 384). Furthermore, we applied subgroup analysis to address potential residential confounding by sex, age, income, and exercise. Female sex is a significant factor for osteoporosis risk, so it was important to assess sex subgroup estimates in this research. Potential misclassification bias may exist due to the spatial unit of exposure assessment as well (i.e., district level), but all air pollutants in this research had the same issue. Given that this study was limited to urban metropolitan cities and randomly sampled participants in the national insurance system, we are not concerned about discernible residual confounding for SO2. However, we added new text in the Discussion that further work on the physiological mechanisms for air pollutants including SO2 is warranted (lines 379-381).

  1. The lack of PM2.5 measurements is a major limitation (and may be tied to the findings with SO2 and no other pollutants) The authors state that widespread monitoring of PM2.5 did not occur until 2013. Is there data available in a subset of the cities prior to this time that would enable some evaluation of PM2.5 exposure?

A: We agree that investigation of PM2.5 for bone fractures is an important research topic. It is possible that long-term exposure to PM2.5 may be associated with osteoporotic fracture. PM2.5 data were available for three metropolitan cities (Seoul, Busan, and Incheon) since 2008. However, the start year of monitoring for PM2.5 also varied among the monitors in these cities, meaning not all monitors had data from 2008 onwards. We purposely did not conduct sub-analysis using PM2.5 for select cities with a limited exposure period as such approaches may lead to selection bias or selective reporting bias. The comparison of the risks from PM10and PM2.5 from different study regions and study periods cannot distinguish the difference in risks differences in particle size and toxicity of chemicals from the heterogeneity of study period, study regions, and study population. We contemplate that a comparison between PM10 and PM2.5 with the same study period and population deserves a single future research article that better explores these complex pollutants, addressing issues such as different timeframe of monitoring, locations of monitoring, chemical composition, etc. We share a reference of Sung, Jung Hun, et al. "Association of air pollution with osteoporotic fracture risk among women over 50 years of age." Journal of Bone and Mineral Metabolism 38 (2020): 839-847. This study used the PM2.5 in the 3 metropolitan cities in South Korea and applied a time-invariant variable for PM2.5 as the PM2.5 estimates were through 2008-2009. We plan to perform research examining longer exposure periods and time-variant exposure measurements for all seven metropolitan cities in South Korea so that the comparison between PM2.5 and PM10 can be examined. To address this comment, we added text to the Discussion section to highlight the need for future work in this area (Lines 459-464).

  1. The exact time period for the analyses is inconsistently described. If 2015 is end of follow-up (Line 110 and 216), then why not include exposure information through at least 2014? Otherwise the different exposure period choices (1-year no lag, 2-year no lag, 3-year no lag, and 2-year with 1 year lag) will have different samples. For example, how is the 1-year no lag exposure calculated with 2015 follow-up data if exposure data are only used through 2013 and not 2014? This may just be an issue of inconsistent description, but if it is in fact an issue of different samples then I suggest the analyses be re-run all using the same dataset (i.e., expand the time range of exposure data collected).

A: We are sorry for this confusion. This was a mistake in the description and there was no issue with different sampling. We considered exposure to air pollution during 2002-2015 but overlooked a typo of 2002-2013 in some places in the manuscript by mistake. This was because the follow-up period was extended from 2002-2013 to 2002-2015 in our study as the new version of the National Health Insurance Service-National Sample Cohort with extended follow-up years became available during our research. We updated the analysis to include the new years but mistakenly neglected to update this text throughout the paper. We apologize for this error. We corrected the period for exposure assessment and follow-up in line 167, line 230, and line 501.

  1. Additional information on the exposure assessment would be helpful for understanding the potential for exposure measurement error. How large (km^2) is the typical district? How many monitors are in the typical district?

A: Each district had average 2.4 monitoring stations (minimum = 1, maximum = 7, Q1 = 2, Q3 = 4). The average size of study districts was 73.1 km2 (minimum 2.8, maximum 753.3, Q1 = 17.9, Q3 = 68.2). We added new text for this information in the Methods (lines 171-173).

  1. None of the effect modification comparisons in Table 3 are statistically significant, so I don’t think the authors should highlight specific estimates from that analysis as statistically significant. In other words, since “there was no significant differences in the HRs among subgroups for all air pollutants” (Lines 294-295), then don’t say that the differences are significant: “estimated impact of exposure to 3-year moving annual average SO2 was significant in women...those age 50-69y…women aged 50-69y..” (lines 289-292).

A: We omitted lines 297-302 describing HRs of subgroups for SO2 (lines 305-309).

Minor/Editorial

  1. What exposure window is used in the Table S3 analyses? And Tables S4 to S8?

A:  In Table S3-S8, 3-year moving annual average concentration of air pollutants was applied. We revised the title of these tables to include ‘3-year moving annual average’ to clarify the exposure window (lines 497-506 in the main text; titles in the supplementary file).

  1. I suggest presenting the justification for 3-year window (Lines 346-350) earlier in the manuscript when first presenting the choice of exposure periods.

A: Lines 346-350 presenting the reasons for choosing exposure up to 3 years were moved to lines 241-245 in the Methods.

  1. In the abstract, what is “central” HR?

A: The word ‘central estimate of effect’ is often used to refer to the main effect estimate other than significance levels (95% CIs). We used this word for O3 as the 95% confidence intervals did not indicate significant association and only the HR of O3 indicated somewhat potentially positive associations.  In this case, we present the full numerical results as well to avoid misinterpretation (line 29 in the abstract).

  1. Line 28 in abstract: subscript O3

A: The number 3 was changed to a subscript.

  1. Line 113: NHIS not NIHS

A: The typo was corrected.

  1. Line 176: “frequency of physical exercise per week” is in days?

A: The original dataset uses the term ‘times’ so we added ‘times’ in line 186.

  1. I suggest being consistent about ppm or ppb for SO2.

A: The unit of SO2 was changed from ppm to ppb throughout the manuscript.

  1. Line 388 has a double negative.

A: We changed the sentence to “On the other hand, our study using the fixed-location measurements did not have information of spatial heterogeneity patterns of O3 within a district.” (line 407).

  1. Table 3 is difficult to read. The horizontal lines are inconsistent and the p-values are split onto multiple lines.

A: We found that Table 3 did not upload correctly during the file uploading process, for which we were not given a confirmation step before completing the submission. We revised the table and apologize for this issue.

Round 2

Reviewer 2 Report

I have no additional comments at this time.